# Parenting a Child with a Neurodevelopmental Disorder during the Early Stage of the COVID-19 Pandemic: Quantitative and Qualitative Cross-Cultural Findings

**DOI:** 10.3390/ijerph20010499

**Published:** 2022-12-28

**Authors:** Noemi Mazzoni, Arianna Bentenuto, Fabio Filosofi, Angela Tardivo, Lane Strathearn, Kasra Zarei, Simona De Falco, Paola Venuti, Giuseppe Iandolo, Michele Giannotti

**Affiliations:** 1Laboratory of Observation, Diagnosis and Educational (ODFLAB), Department of Psychology and Cognitive Science, University of Trento, 38068 Rovereto, Trento, Italy; 2Observation and Functional Diagnosis Division, PSISE Clinical and Developmental Psychological Service, Calle Albendiego 7, 28029 Madrid, Spain; 3Department of Pediatrics, Carver College of Medicine, University of Iowa, Iowa City, IA 52242, USA; 4Center for Disabilities and Development, University of Iowa Stead Family Children’s Hospital, Iowa City, IA 52242, USA; 5Hawkeye Intellectual and Developmental Disabilities Research Center (Hawk-IDDRC), University of Iowa, Iowa City, IA 52242, USA; 6Department of Psychology, School of Biomedical Sciences, European University of Madrid, Calle Tajo S/N, (Urb. El Bosque), Villaviciosa de Odón (Madrid), 28670 Madrid, Spain

**Keywords:** mental health, parents, neurodevelopmental disorder, autism spectrum disorder, parental stress, externalizing behaviors, cross-cultural

## Abstract

Research during the COVID-19 pandemic has shown a strong relationship between child symptoms, parental stress, and mental health challenges. The pandemic has changed family routines, worsening child symptomatology and parental burden. The aim of this study was to investigate how the magnitude of the perceived changes in child externalizing behavior, parental stress, and discontinuity of therapy—from before to during the COVID-19 pandemic—affected parental mental health during the pandemic. Moreover, we sought to compare these aspects cross-culturally between European countries and the USA. To these purposes, we asked Italian, Spanish, and U.S. parents of children with neurodevelopmental disabilities (NDD) to complete an online survey. Quantitative results showed that increased parental stress may have contributed to a worsening in parental psychological distress, regardless of culture. Moreover, they suggested an indirect effect of child externalizing behaviors on parents’ psychological distress via parental stress. Qualitative analyses highlighted that the lack, or discontinuity, of therapeutic activities may have been one of the key contributors to parenting burden during the COVID-19 pandemic. Finally, qualitative results highlighted resilience factors that could have decreased the risk of psychological problems during the pandemic, such as a strong sense of parental efficacy and the ability to adapt to changing family dynamics.

## 1. Introduction

Rearing children and adolescents with neurodevelopmental disorders (NDD) is often associated with elevated parental stress [1,2,3] and increased risk of mental health problems, such as anxiety and depression [4,5,6,7,8], especially in parents of children with autism spectrum disorder (ASD) [9,10,11]. Evidence suggests that parental stress is driven by child behavioral difficulties [3,12,13,14]. Specifically, research predating the COVID-19 pandemic showed that the severity of child externalizing behaviors increases the demand of child care and parental psychological distress [1,15]. Overall, these studies showed that the severity of child symptoms and behavioral challenges was significantly associated with parental stress and mental health problems [8,11,16,17,18]. Moreover, the association between child symptomatology and parental psychological distress is mediated by several factors, including concern for the future, parental stress, marital conflict, and the family’s economic situation [4,14,19].

Notably, the COVID-19 pandemic and the measures adopted to contain the spread of the virus may have exacerbated these factors and further increased parenting demands [20,21,22,23,24]. Critically, the measures taken to contain the spread of COVID-19 have abruptly challenged the daily routines of families of NDD children and disrupted the provision of therapeutic and educational services [25,26,27], potentially leading to increased child externalizing behavior and parental stress [28]. In this regard, it is recognized that children and adolescents with NDD benefit from stable routines [29] that include scholastic, therapeutic, and sport/leisure activities. Family support represents another important factor for managing the parental burden [30,31,32], especially during stressful circumstances such as the case of the COVID-19 pandemic [25,26,27,33]. Indeed, having a strong familiar and social support, either formal (e.g., professional services and community resources) or informal (e.g., friends, peer groups, school or work settings, online communities), have been shown to have a positive effect on parental stress and parents’ wellbeing [34] and contribute to increased family resilience and adaptation when facing overwhelming situation (e.g., crisis or disaster, such as that which the worldwide pandemic was) [35]. Recent findings showed that the decrease in therapeutic/rehabilitation support during COVID-19 home confinement predicted higher externalizing behaviors in children with NDD [36], which in turn were associated with greater parental stress [28]. These studies underline the importance of considering the strong association between children and parents’ adjustment when the psychological impact of the COVID-19 pandemic on NDD families is investigated. In sum, research during the COVID-19 pandemic has shown a strong relation between child symptoms, parental stress, and mental health outcomes. However, it is unclear how the magnitude of the perceived changes in child externalizing behavior, parental stress, and discontinuity of therapy—from before to during the COVID-19 pandemic—affected parental mental health during the early stage of the pandemic.

Although restrictive measures around the world were modulated at different times, according to fluctuating COVID-19 conditions, they generated similar negative psychological consequences, especially for individuals with special needs and their families [25,28,37,38]. Interestingly, the negative impact of COVID-19 on the psychological well-being of parents of children with NDD seems to be similar in different cultures, with child maladaptive behavior associated with increased parental psychological distress [39,40,41]. However, a direct cross-cultural comparison between European countries and the USA is lacking.

To fill these gaps, we asked Italian, Spanish, and U.S. parents of children with NDD to complete an online survey referring to the period before and during the COVID-19 pandemic. The first aim of this study was to examine the predictors of parents’ psychological distress, including culture, pandemic-related variables, perceived changes in child externalizing behaviors, and parental stress and therapy. The second aim of this study was to further explore, using qualitative methods, the most salient aspects of parental experiences during the early stage of the COVID-19 pandemic in the context of NDD. Indeed, recent research has shown that qualitative analyses can help to unveil the emotional, social, and health challenges associated to raising a child with NDD [42,43].

## 2. Materials and Method

### 2.1. Participants

Only parents over eighteen years of age, with at least one child aged between 3 and 17 years, could participate in the study. This study is part of a larger project on parenting during the COVID-19 pandemic in different countries, which involved a total of 1494 parents (723 Italian, 530 Spanish, and 243 U.S. parents). Of these, the parents of children with special needs were N = 107 for the Italian sample, N = 50 for the Spanish sample, and N = 48 for the U.S. sample. From this larger sample, we extracted the data relative to parents of children with NDD that are presented in the present study. We thus excluded all the parents of children with psychiatric disorders, general medical conditions, or any other form of disability that was different from NDD, so that the final study sample included 143 parents of children with a neurodevelopmental disorder, of which 82 were from Italy (59.4%), 27 from Spain (18.9%), and 31 from the United States (21.7%). The majority of children had a diagnosis of ASD (*n* = 68), followed by attention deficit and hyperactivity disorder (ADHD; *n* = 21), language disorder (LD; *n* = 17), specific learning disabilities (SLD; *n* = 15), intellectual disability (ID; *n* = 9), and other neurodevelopmental conditions (e.g., Down syndrome; *n* = 13). Characteristics of the study participants by country are displayed in Table 1.

### 2.2. Measures

#### 2.2.1. Socio-Demographic and Pandemic-Related Variables

We collected sociodemographic data, including parental employment status and family socioeconomic status (SES). The survey also included additional questions about pandemic-related contextual factors, such as having or knowing someone who tested positive for COVID-19 (experience of infection; “Yes” or “No”) and the amount of time that the child was followed by therapists and/or rehabilitation specialists before and during the COVID-19 lockdown (Δ therapy). Specifically, respondents were asked to rate on a 5-point Likert scale (from 1 = “Much less than before” to 5 = “Much more than before”) the amount of therapy/rehabilitation received by their child at the time of completion compared to the period before the COVID-19 pandemic.

#### 2.2.2. Psychological Distress

We used the General Health Questionnaire-12 (GHQ-12) [44,45,46], a widely-used questionnaire for the assessment of psychological distress. It comprised 12 items rated on a four-point scale (i.e., “less than usual”, “no more than usual”, “rather more than usual”, or “much more than usual”) investigating perceived changes in symptoms or behaviors during the last two weeks. In this study, we adopted the Likert scoring method (0-1-2-3). The GHQ-12 has previously shown strong psychometric properties [23,24], with Cronbach’s alpha coefficients ranging from 0.798 (USA) to 0.819 (Italy) in our sample (specifically, Italy = 0.81, Spain = 0.799, USA = 0.798).

#### 2.2.3. Child Externalizing Behavior

The Strengths and Difficulties Questionnaire (SDQ) [47,48] has been used to assess mental health in children aged between 3 and 17 years. For this study, we used the externalizing score (SDQext; ranging from 0 to 20) generated from the sum of the five “hyperactivity/inattention” items and the five “conduct problems” items. Participants were asked to rate their response on a three-point Likert scale (0 = “not true”; 1 = “somewhat true”; 2 = “completely true”). The questionnaire has previously demonstrated good psychometric properties [25,26], with Cronbach’s alpha values ranging from 0.701 (Italy) to 0.882 (USA) in our sample (specifically, Cronbach’s alpha pre-COVID-19: Italy = 0.701, Spain = 0.730, USA = 0.845; Cronbach’s alpha during COVID-19: Italy = 0.769, Spain = 0.715, USA = 0.882).

#### 2.2.4. Parental Stress

Ten items of the Parental Stress Scale (PSS) [49] were selected to assess feelings and perceptions related to parental role and experiences. Participants were asked to rate their responses on a 5-point Likert scale (1 = “strongly disagree”, 5 = “strongly agree”; range 10–50). Since the scale was not available in Italian, a back-translation procedure (S.D.; M.G.; N.M.) was performed. The full version of the PSS has previously revealed satisfactory psychometric properties [49,50]. In the current study, the brief scale showed acceptable internal consistency, with Cronbach’s alpha values ranging from 0.724 (Spain) to 0.873 (USA) (specifically, Cronbach’s alpha pre-COVID-19: Italy = 0.788, Spain = 0.738, USA = 0.842; Cronbach’s alpha during COVID-19: Italy = 0.805, Spain = 0.724, USA = 0.873).

#### 2.2.5. Open Question on Parenting during the COVID-19 Pandemic

Participants were asked to freely describe the most significant aspects of their experience as a parent during the COVID-19 pandemic through an optional open question at the end of the questionnaire.

### 2.3. Procedures

Data were gathered using a web-based cross-sectional anonymous survey on parenting during the first wave of the COVID-19 pandemic (March–November 2020) using Qualtrics software [51]. We recruited participants using a snowball sampling technique and promoting the study on the universities’ (masked for review) mailing list and social media. Information on study aims, methods, and privacy was described in the first section of the survey. All the participants volunteered, and informed consent was obtained before commencing the survey. Notably, each item of the PSS and SDQ was repeated on two occasions. On the first occasion (T1), participants were asked to refer to the present, while on the second occasion (T0), they were asked to refer to a month before the COVID-19 pandemic. Survey completion took around 15 min. Only parents who completed the questionnaire in more than 500 s were included in the study. This study is part of a larger project on parenting during the COVID-19 pandemic in different countries, and it extended previous findings on Italian families of children with NDD [28]. This research project was approved by the IRB of the (masked for review).

### 2.4. Statistical Analysis

Firstly, we checked for potential differences between countries on control variables such as parents’ age, family SES, employment status, child gender, and age and experience of contagion by using the MANOVA test for continuous variables with the Bonferroni post hoc test, and the chi-squared test for categorical variables. Next, we subtracted the total score at T0 (before the pandemic) from the total score total at T1 (during the pandemic) for both SDQext and PSS scores to generate two variables reflecting the perceived changes on child externalizing symptoms (ΔSDQext) and parental stress (ΔPSS). For the first study aim, we performed a hierarchical linear regression to investigate the predictors of psychological distress (dependent variable) during the first wave of the COVID-19 pandemic in parents of children with NDD, considering the role of culture and sociodemographic and pandemic-related factors, as well as perceived changes in child externalizing behaviors, parental stress, and therapy/rehabilitation received by children. To allow this, we first included two dummy variables in the analysis indicating whether the parent was (i) from Italy, (ii) from Spain, or (iii) from the USA. In the second block, we entered socio-demographic and pandemic-related variables, including parent’s age, employment status, SES, and experience of contagion. Next, perceived changes in child externalizing behaviors from before to during the COVID-19 pandemic (ΔSDQext) and Δtherapy were added in the third block. Finally, ΔPSS was introduced in the fourth block of the regression model. The variance inflation factors (VIFs) were examined to detect multicollinearity among the predictors [29], with values smaller than 4 considered as acceptable [30]. The statistical analyses were conducted using the software SPSS version 25.0 for Windows [52] (IBM Corp).

### 2.5. Open Question

At the end of the survey, participants were asked to freely describe the most significant aspect of their parenting experience during the early stage of the COVID-19 pandemic through an open question. Specifically, the open question was “What aspect of your experience as a parent in this particular time do you think is more significant?” (in Italian “Qual è l’aspetto che ritiene più significativo della sua esperienza come genitore in questo particolare momento?”, in Spanish “¿Cuál es el aspecto más significativo de su experiencia como padre en este momento en particular?”). This part was not mandatory, and parents were free to answer by sharing with us their thoughts about their experience as a parent during the pandemic, or to skip the question and terminate the survey.

## 3. Results

### 3.1. Quantitative Results on Predictors of Parental Psychological Distress

Descriptive statistics are displayed in Table 2 (bivariate correlations are displayed in Appendix A).

The MANOVA showed significant differences between countries (*F*(4, 136) = 6.29, *p* < 0.001; *Wilk’s Λ* = 0.710, *partial η*2 = 0.15). Univariate tests revealed statistically significant differences between groups on child age (*F*(3,140) = 3.15, *p* < 0.05) and family socioeconomic status (*F*(3, 140) = 16.17, *p* <0.001), while no differences emerged for parent’s age and Δtherapy (*p* > 0.05). Specifically, Bonferroni post hoc analyses showed that Italian parents showed lower SES than American parents (*p* < 0.001) and higher levels than Spanish parents (*p* = 0.03). With respect to age, American children were older than Italian children (*p* = 0.04) but were not significantly different from the Spanish group (*p* > 0.05).

Regarding our first study aim, the overall regression model on parental psychological distress (Table 3) was statistically significant (*F*(9, 140) = 4.88, *p* < 0.001, *R*^2^ = 0.25). In the first block, the variable country was not significant and explained only 2% of the variance (*F*(2, 140) = 4.88, *p* = 0.087). Similarly, sociodemographic variables (parent age, current employment status, SES) and experience of contagion were included in the second block of the hierarchical regression but did not contribute significantly to parental psychological distress (*F*(6, 140) = 4.88, *p* = 0.087, *R*^2^
*change* = 0.05, *p* < 0.13). Next, the variables we entered in the third block added a considerable amount of variance (*R*^2^
*change* = 0.11, *p* < 0.001), showing a statistically significant model (*F*(8, 140) = 4.08, *p* <0.001, *R*^2^= 0.19, *p* < 0.001). In particular, changes in child externalizing behaviors as perceived by the parent (ΔSDQext) contributed significantly to psychological distress (*β* = 0.30, *p* < 0.001), while no effect of Δtherapy was found (*β* = −0.12, *p* < 0.13). In the last block, perceived changes in parental stress (ΔPSS) were entered as explanatory variables, adding a significant contribution to the final model (*R*^2^
*change* = 0.05, *p* = 0.003). In particular, change in parental stress (ΔPSS) was the only variable that significantly predicted mental health in parents (*β* = 0.291, *p* = 0.003), with increased parental stress during the COVID-19 pandemic predicting higher scores of the dependent variable (GHQ). In contrast to the previous block, ΔSDQext was no longer associated with psychological distress (*β* = 0.137, *p* = 0.14). Finally, Δtherapy showed only a marginal tendency toward significance (*β* = 0.14, *p* = 0.07), with a decrease in therapy/rehabilitation activities during the COVID-19 pandemic associated with higher psychological distress in parents of children with NDD.

### 3.2. Qualitative Results on Parenting a Child with NDD during the Pandemic

In total, 91 out of 143 parents (63.63%) with children with NDD replied to the open question. We analyzed the open questions using qualitative content analysis methodology [53,54]. Specifically, we first extrapolated 12 distinct thematic units from the 91 responses. Successively, we aggregated the 12 theme units into 4 thematic macro-categories: (1) opportunities; (2) adaptive attitudes; (3) lack of support; (4) psychological and relational difficulties. The first macro-category “opportunities” counted 28 occurrences among the 91 responses and included references to the opportunity to spend time with the child and strengthen the parent–child relationship. Specifically, the following thematic units were included: (a) opportunity to have more time to spend with the child (*n* = 26; 28.57% of respondents); (b) experience of parental effectiveness (*n* = 11; 12.09 %). The second macro-category “adaptive attitudes” counted 29 occurrences among the 91 responses and included references to ability to adapt to relationship dynamics during the pandemic restrictions. In particular, the following themes emerged: (a) adaptation to change and flexibility (*n* = 15; 16.48% of respondents); (b) being patient (*n* = 8; 8.79%); (c) to convey calm/protection (*n* = 13; 14.28%). The third macro-category “lack of support” occurred 45 times among the 91 responses and included references to the difficulties in managing school and distance learning. The thematic units that emerged were (a) difficulties with school and distance learning (*n* = 25; 27.47%); (b) lack of therapeutic support (*n* = 14; 15.38%); (c) combining different roles (*n* = 26; 28.57%); and (d) lack of child’s extracurricular experiences and social interactions (*n* = 8; 8.79%). The fourth macro-category “psychological and relational difficulties” occurred 34 times among the 91 responses and included references to emotional states during the lockdown period such as (a) stress and negative feelings (*n* = 20; 21.98%); (b) concern for the child’s well-being (*n* = 11; 12.09%); (c) inadequacy and sense of helplessness (*n* = 11; 12.09%).

Through the analysis of qualitative results based on occurrence [53], we observed overall that the “lack of support” macro-category had the greatest number of occurrences compared to the other categories. In addition, looking at the occurrence of the basic theme units, we noticed that the “opportunity to have more time to dedicate to the child” and “combining different roles” macro-categories were the most frequent units (28.57% of respondents) in parents’ replies, followed by “difficulties with school and distance learning” (27.47% respondents). Furthermore, we analyzed the parents’ answers at a cross-cultural level. When the occurrences were compared between the three countries, both some differences and commonalities emerged in the frequencies of the four macro-categories (Table 4). For instance, the Italian parents, on the one end, described the pandemic period as an opportunity to spend time with the child and strengthen the parent–child relationship (*n* = 22; 44.00%) and, on the other end, they underlined the perception of lack of support received (*n* = 22; 44.00%). With respect to Spanish parents, their reported most frequently the presence of adaptive attitudes (*n* = 9%; 47.37%), as well as psychological and relational difficulties (*n* = 8; 42.10%) and the lack of support (*n* = 7; 36.84%). Finally, the aspects reported more frequently by the American parents were the lack of support (*n* = 16; 72.73%) and psychological and relational difficulties (*n* = 14; 63.64%).

## 4. Discussion

Since rearing a child with a NDD is also associated with elevated parental strain during normal circumstances, the dramatic changes in families’ routines that occurred during the early stages of the COVID-19 pandemic may have increased the risk of parents experiencing higher levels of psychological distress. In this regard, the aim of this study was to shed light on parenting experiences in families of children with NDDs during the pandemic, comparing European and U.S. families. Specifically, using a quantitative and qualitative approach, we examined (a) how perceived changes in different psychological dimensions, from before to during the COVID-19 pandemic, affected the mental health of parents, and (b) the most significant aspect that characterized parenting experience in the early pandemic period.

Overall, our findings showed that an increase in child externalizing behaviors and parental stress as perceived by the parent contributed to a worsening in parental psychological distress, regardless of culture, sociodemographic factors (parents’ age, employment status, family SES), and experience of contagion.

In this study, we measured the externalizing behaviors as perceived by the parents. During lockdown, most of the parents spent a greater amount of time with their children, compared to the period before the pandemic, serving as teachers and educators (besides their roles as parents). A recent study found that some abilities linked to externalizing behaviors (such as self-monitoring, emotional control, and emotional regulation) were perceived as worse at school than in family environments [55]. Therefore, it could be argued that the parents overrated their children externalizing behaviors during the home confinement and that the perceived worsening of the externalizing behaviors was partially associated to the fact that, during the lockdown, the scholastic activities were realized at home by the parents. On the other hand, the urgency of caring and rearing children with NDD without the usual external support may have led to overwhelm and increased parental stress, which in turn may have biased the judgement of children’s externalizing behaviors. However, independently from the reasons why the externalizing behaviors were perceived as worsened, it is important to carefully consider the parents’ perceptions, as there is a well-documented reciprocal influence between the severity of child behavioral difficulties and caregiver burden [20,56,57]. Our results corroborate this relation. Indeed, we found that changes in parental stress constituted a remarkable predictor of mental health, in line with previous research documenting its robust effect during pandemic circumstances in parents of children with typical and atypical development [13,28,58,59]. Moreover, we found that the influence of externalizing behaviors was no longer significant when parenting stress was entered in the regression model, suggesting an indirect effect of perceived changes in child externalizing behaviors on psychological distress via perceived changes in parental stress. This confirmed the well-known link between the severity of child behavioral difficulties (e.g., hyperactivity, conduct problems) and caregiver burden also during the pandemic [12,13,60]. Contrary to our hypothesis, we did not find a significant direct link between reduced therapy received by the child and parental psychological distress. However, there was a clear tendency toward significance, suggesting that the decrease in therapeutic/rehabilitation support may contribute to determining parental mental health. We acknowledge that the limited sample size is a major study limit that might have prevented some results from reaching significance, such as the effect of reduced therapies. An increased sample size may have clarified this effect and is thus recommended for future research. Interestingly, our qualitative results support the existence of a relation between therapy reduction and parental psychological distress, showing that discontinuity and lack of therapeutic activities emerged as one of the key aspects related to parenting experience during the COVID-19 pandemic. These findings are consistent with recent studies, revealing the adverse effects of service disruption/difficult or delayed transition to telehealth for children with developmental disabilities and their families [26,27,28].

Results of qualitative analyses also pointed out some additional risks related to increased parental burden during the pandemic, as well as some resilience factors. Focusing on the risks, the lack of educational support was reported as the greatest issue by parents that negatively contributed to NDD child adjustment and parental psychological well-being. Additionally, other concerns reported in our survey by the parents included the difficulties in combining different roles and in managing distance learning education for children. The abrupt onset of needing to combine roles as parent, teacher, and therapist of their own children, imposed by COVID-19 restrictions, with little or no support by professionals and schools, may have significantly challenged family dynamics, increasing parental burden and feelings of exhaustion. This suggests that difficulties in balancing work and child rearing may have amplified demands on parents during the pandemic, constituting a significant source of parental distress and worsening parental psychological health.

Notably, besides the difficulties related to parenting NDD children while COVID-19 stay-at-home restrictions were in place, the open question on our survey also highlighted some strengths and new opportunities related to family adjustment. Specifically, parents appreciated the opportunity to spend more time with their children and reported a strengthened parent–child relationship, as well as greater patience and flexibility, in line with previous qualitative findings in the Spanish [61] and Italian [28] contexts. These results suggest that a high sense of efficacy as a parent and good parental ability to adapt to changed family dynamics could be pivotal resilience factors that decrease the risk of psychological problems during pandemic times.

When qualitative data were compared between the three countries, other interesting information emerged. We found that that the lack of support was a common issue reported by the parents of all the three countries, although with different percentage of frequency (i.e., higher in the USA than in Italy and Spain). Moreover, Spanish and Italian parents reported a family-wise positive aspect of spending more time at home with the children, while this was less present in U.S. parents’ answers. This may have been due in part to cultural differences and in part to differences in health and social policy directed to children with NDD and their families.

For instance, in Italy and Spain, the National Health Service is public, founded by the state, and freely accessible to all citizens without limitation based on income, gender, or age. Among the public services provided by the state, the Neuropsychiatry Service (Italy) and Hospitals, Health Centers, and the Psycho-Pedagogical Guidance Teams (Spain) are deputed to provide assessment for children from 0 to 18 years of age with developmental or psychiatric disorders.

In Italy, in addition to the assessment, the public Neuropsychiatry Service provides professional health services such as psychological, neuropsychological, speech, and psychomotor therapies (e.g., in Italy, for children with NDD, the state freely provides 1 h per week of psychomotor therapy and 1 h per week of speech and language therapy). Italian families can access all these services for free or at a minimal cost after a request made by the pediatrician (who cares for children from 0 to 14 years at no cost).

In Spain, according to the degree of dependency validated by the Regional Coordination Centers for assessment after the assessment, the child can access a service paycheck that the family can use with public and private convention socio-health centers.

Besides the public Neuropsychiatry (Italy), Hospitals, Health Centers, and Psycho-Pedagogical Guidance Teams (Spain), the same psychological, neuropsychological, speech, and psychomotor services can be provided privately. The quality and quantity of private institutes and professionals vary among the country regions. During the COVID-19 lockdown, all the in-presence therapies, health, and educational activities were suspended in the public and private sectors. Public services, private centers, and professionals activated alternative services (e.g., telerehabilitation) [62,63,64]. For instance, in some Italian regions (e.g., Campania, Lombardy, Marche, Emilia Romagna), individual state-funded services provided online intervention and parental support. However, there was significant variability regarding when these alternative services were activated—if at all—and the type of activities provided. They were not decided at a national level but activated individually by public and private centers. Hence, there was neither homogeneity between the services delivered by the different neuropsychiatric units across the country nor between the ones provided by private professionals. This heterogeneous situation was similar in Italy and Spain.

Similarities between Italy and Spain also exist in the educational system. Education is free for all children and adolescents and accessible to all. However, there is a difference: while in Spain, the children with special needs attend special schools, in Italy, there is no difference in the type of school attended by the children with and without special needs. In Italy, children are included in the same class of children with typical development (TD) with one-to-one teaching support (freely provided by the state). This fosters the inclusion of NDD children among their TD peers, helps reduce stigma and discrimination, facilitates social interactions, and promotes support and collaboration between NDD and TD children and their families.

In the USA, the situation is different. Children with NDD and their families may face limited access to equal opportunities in education and health. This may happen because high-quality public education, health, and social care services offered by the state are not universal or comprehensive of all the needed services, and thus not always able to respond adequately to the complex needs of NDD children and their families. The access to the needed health and educational services may be expensive. This may represent a barrier for families with financial difficulties and generate disparities between low- and high-income families, which may have been even exacerbated by the suspension of economic activities and/or loss of job experienced during the pandemic. Moreover, as in Spain, in the USA, the education of children with special needs is not inclusive but realized in special schools. In such a scenario, the financial demanding as well as the emotional and social challenges associated with raising a child with NDD could be worsened compared to that experienced by families who live in a society where the state provides adequate services at a minimal cost.

Our qualitative data seem to reflect these differences in health and education systems. On one hand, the impossibility of assessing in presence therapies and the variability regarding the time and the quality of the alternative services activation—although essential to support the families and the children during the lockdown were perceived as less effective than in presence ones—may have generated the feeling of “lack of support” that emerged in all the three countries. On the other hand, the feeling of being part of an inclusive scholastic community may have allowed for the maintenance of contact (although online) with the other families and child peers and, hence, be reflected in less frequent reports of psychological and relational difficulties. Moreover, the possibility of accessing the school freely and the child treatment and health services at no or minimal cost may have contributed to reducing parents’ concerns—and, in turn, parental stress and overwhelm—in the cases of financial difficulties/job loss due to the pandemic. Furthermore, the presence of a schoolteacher dedicated to the student with special needs provided by the state, who assisted the student individually during the lectures and prepared adapted materials, may have supported the parents during the online teaching. All these aspects pertain more to Italian and Spanish societies than to the U.S. one and may be the reasons why U.S. parents more frequently reported feelings of a “lack of support” and a less “positive” experience related to the lockdown, such as the joy of spending more time with the children or the ability to convey to them calmness and protection.

Besides these aspects, also cultural characteristics such as individualism/collectivism dimensions [65] have been shown to play a role in driving pandemic responses cross-culturally [66,67,68]. The USA is classified as having the culture with the world’s highest level of individualism, with a prioritization of individual freedom, autonomy, and personal interest over collective wellbeing [69], and Italy’s and Spain’s cultures are also characterized by some degree of individualism. Interestingly, research on the effect of cultural dimensions during the pandemic has shown that in the cultures characterized by individualism, the promotion and the adherence to risk mitigation behaviors, such as social distancing, vaccination, and mask wearing, have been more difficult and less effective compared to collectivistic cultures, which instead displayed more adaptive responses [67,70]. Cultures with high individualism experienced higher infection cases and deaths than collectivist cultures [69,70,71]. For instance, the USA recorded the largest number of infected cases and number of deaths due to COVID-19, and Italy and Spain were among the most severely affected countries in Europe. Moreover, people living in individualistic cultures were more resistant to altering their behaviors according to the lockdown measures disposed by governments [67] and experienced higher levels of stress and isolation [72]. Therefore, in such a scenario, it is possible that the cross-cultural differences that emerged from our qualitative analyses may be partially attributable to the degree of cultural individualism and partially to the different health, social, and education systems.

### Clinical and Social Implication

Altogether, our quantitative and qualitative results have relevance at a clinical level as they suggest the importance of ensuring continuity of care for children with NDD (e.g., by activating timely telehealth programs) that can support parents and limit the deterioration of parental psychological health when in-person activities are restricted. The imposed social distance has challenged community and professional networks so that families of children with NDD may feel even more vulnerable and isolated because of a lack of social and therapeutic support. Thus, clinical activities for children with NDD should be rapidly adapted during pandemic times in order to respond to patients’ and their families’ needs. The possibility of using technology to initiate and manage interventions could provide psychological support to all family members and should be a priority when, for any reason, in-person activities are restricted. Consistently, a growing number of studies promote the utility and feasibility of a telehealth approach to interventions for children with NDD [26,64,73] that also appear to be feasible in developing countries [74]. In this scenario, parent-mediated interventions may facilitate parent-child relations, promote parental empowerment, and reduce parenting stress [27,75,76]. Other important aspects that should be considered in order to prevent/reduce the feeling of “lack of support” and to mitigate the NDD parents overwhelming are (1) fostering the accessibility of a high level of health and educational services at minimum costs, especially for families with financial difficulties and low incomes; (2) promoting inclusion and reducing stigma and isolation of children with NDD and their families by fostering the creation of good network of social support.

## 5. Conclusions and Limitations

In sum, our findings showed that, in all the countries analyzed in this study, the parental perception of worsened parental stress and child externalizing behaviors (from before to during COVID-19) negatively impacted parents’ general mental health. This could plausibly be due to abrupt routine changes derived from COVID-19 restrictions that resulted in decreased social and professional services and produced more frequent child externalizing behaviors [77,78], which in turn increased the parents’ psychological distress via parental stress. As a clinical implication, our data highlight the urgency of paying attention to psychological wellbeing in families of children with NDD that experience home confinement and to activate/adapt clinical interventions (e.g., using telehealth) to efficiently provide adequate healthcare services. Finally, in addition to the negative effects, our results highlighted resilience factors—such as a high sense of efficacy as a parent and good parental ability to adapt to changed family dynamics—that could decrease the risk of psychological problems during pandemics such as COVID-19.

Some study limits need to be acknowledged. Limited sample size is the major limit of this study and thus caution should be taken in driving firm conclusions from our results. Future research with a greater sample size is needed to corroborate our findings. Moreover, the limited sample size might have prevented some results from reaching significance, such as the effect of reduced therapies. In this sense, our qualitative results enrich quantitative findings and are important to better understand the challenges experienced by parents of children with NDD during this burden period and the factors that contributed to their overwhelming.

Another limitation is relative to the measure of children externalizing behaviors, which was not directly collected through children observation, due to the impossibility of in-presence activities, but was collected indirectly by parents’ perception. Thus, given the higher parental stress level during COVID-19 home confinement, we cannot exclude that parents may have overreported behavioral difficulties in their children. Moreover, recent findings showed that life satisfaction decreased as the days of home confinement progressed [79]. We collected data during the first wave of the pandemic period, which may have been characterized by feelings of uncertainty and greater difficulties in balancing different roles in child-rearing with little or absent external support. Therefore, future research investigating the experience of parents of children with NDD in subsequent periods of the COVID-19 pandemic and with consideration of the home confinement duration are desirable.

In conclusions, despite these limitations, our study provides novel and interesting cross-cultural information about determinants of parental psychological distress in the USA and European countries during COVID-19 lockdowns, highlighting either differences or commonalities that may guide clinical and socio/educational good practice.

## Figures and Tables

**Table 1 ijerph-20-00499-t001:** Characteristics of the study participants by country.

Variables	Italy (*n* = 85)	Spain (*n* = 27)	USA (*n* = 31)	Test	*p*-Value
	M (SD) or *n* (%)	M (SD) or *n* (%)	M (SD) or *n* (%)		
Child age	5.69 (3.79)	6.37 (3.49)	7.65 (3.65)	3.16	*p* = 0.046
Parental age	42.86 (7.39)	41.70 (6.29)	42.32 (5.92)	0.30	*p* = 0.741
Current employment status				28.43	*p* < 0.001
*Working*	42 (49.4%)	11 (40.7%)	31 (100%)		
*Not working*	43 (50.6%)	16 (59.3%)	0 (0%)		
Experience of contagion				2.09	*p* = 0.351
*No*	48 (56.5%)	13 (48.1%)	13 (41.9%)		
*Yes*	37 (43.5%)	14 (51.9%)	18 (58.1%)		
Family SES	34.31 (12.86)	26.61 (14.59)	46.33 (14.15)	16.17	*p* < 0.001
Child gender				8.36	*p* = 0.015
*Male*	67 (78.8%)	18 (66.7%)	16 (51.6%)		
*Female*	18 (21.2%)	9 (33.3%)	15 (48.4%)		
ΔTherapy	1.81 (0.90)	1.78 (0.84)	2.23 (1.08)	2.51	*p* = 0.085

Note: SES: socioeconomic status; Δ: changes from before to during COVID-19 pandemic; *p*-values are referring to MANOVA for continuous variables and chi-squared tests for categorical variables.

**Table 2 ijerph-20-00499-t002:** Descriptive statistics of the Italian, Spanish, and American samples.

Variables	Italy (*n* = 85)		Spain (*n* = 27)		USA (*n* = 31)	
	M (SD)	Min; Max	M (SD)	Min; Max	M (SD)	Min; Max
Pre-COVID-19 SDQ_ext	6.93 (3.14)	3; 17	8.89 (3.50)	2; 15	8.16 (4.18)	0; 20
COVID-19 SDQ_ext	7.75 (3.66)	0; 18	9.67 (3.40)	1; 16	9.06 (4.94)	0; 20
Pre-COVID-19 PSS	25.85 (6.91)	11; 43	26.15 (6.46)	16; 40	26.74 (6.53)	14; 40
COVID-19 PSS	27.91 (7.26)	11; 45	29.15 (6.76)	18; 42	30.74 (7.54)	13; 45
COVID-19 GHQ	21.77 (5.65)	6; 35	24.18 (5.44)	9; 35	22.61 (3.80)	15; 30
ΔSDQ_ext	0.82 (2.30)	−6; 11	0.78 (1.69)	−3; 4	0.90 (2.41)	−4; 6
ΔPSS	2.06 (4.41)	−10; 20	3.00 (4.72)	−5; 15	4.00 (5.02)	−4; 15

Note: SDQ_ext: child externalizing behavior; PSS: parental stress; GHQ: parental mental health; Δ: changes from before to during the COVID-19 pandemic.

**Table 3 ijerph-20-00499-t003:** Hierarchical regression predicting parental psychological distress (GHQ) during the COVID-19 pandemic.

	Covid-19 GHQ							
	Model 1		Model 2		Model 3		Model 4	
Variables	*β*	*CI*	*β*	*CI*	*β*	*CI*	*β*	*CI*
Spain vs. others	−0.117	−4.29; 1.15	−0.047	−3.68;2.43	−0.056	−3.67; 2.18	−0.055	−3.57; 2.10
Italy vs. others	0.092	−1.19; 3.16	0.155	−0.75; 4.07	0.152	−0.68; 3.94	0.113	−1.05; 3.47
Parental age			−0.147	−0.24; 0.01	−0.112	−0.210; 0.037	−0.080	−0.18; 0.05
Current employment status			0.201	−0.05; 4.35	0.137	−0.64; 3.57	0.120	−0.76; 3.33
Family SES			0.049	−0.05; 0.09	0.037	−0.05; 0.08	−0.016	−0.07; 0.06
Experience of contagion			0.086	−0.85; 2.67	0.082	−0.80; 2.53	0.096	−0.60; 2.63
ΔSDQ_ext					0.303 ***	0.34; 1.09	0.137	−0.11; 0.77
ΔTherapy							−0.142	−1.66; 0.08
					−0.124	−1.59; 0.20		
ΔPSS							0.291 **	0.11; 0.54
Δ*R*^2^	0.035		0.050		0.114		0.053	
*R* ^2^	0.035		0.084		0.198		0.251	

Note: *** *p* < 0.001; ** *p* < 0.01; GHQ: parental mental health; SES: socioeconomic status; SDQ_ext: child externalizing behavior; PSS: parental stress; Δ: change from before to during the COVID-19 pandemic.

**Table 4 ijerph-20-00499-t004:** The thematic areas emerged from the qualitative analysis are reported in the first column. For all the thematic areas, the frequencies of the responses are reported either by considering the responses overall (column 2) or separately for each country (columns 3, 4, and 5).

Thematic Areas	*f* (TOT)	*f* (Italy)	*f* (Spain)	*f* (USA)
Opportunities	28 (30.77%)	22 (44%)	4 (21.05%)	2 (9.09%)
- Opportunity to have more time to spend with the child	26 (28.57%)	20 (40%)	4 (21.05%)	2 (9.09%)
- Experience of parental effectiveness	11 (12.09%)	8 (16%)	2 (10.53%)	1 (4.55%)
Adaptive Attitudes	29 (31.87%)	14 (28%)	9 (47.37%)	6 (27.27%)
- Adaptation to change and flexibility	15 (16.48%)	7 (14%)	4 (21.05%)	4 (18.18%)
- Being patient	8 (8.79%)	3 (6%)	4 (21.05%)	1 (4.55%)
- To convey calm/protection	13 (14.28%)	8 (16%)	2 (10.53%)	3 (13.64%)
Lack of Support	45 (49.45%)	22 (44%)	7 (36.84%)	16 (72.73%)
- Difficulties with school and distance learning	25 (27.47%)	12 (24%)	4 (21.05%)	9 (40.91%)
- Lack of therapeutic support	14 (15. 38%)	9 (18%)	1 (5.26%)	4 (18.18%)
- Combining different roles	26 (28.57%)	14 (28%)	3 (15.79%)	9 (40.91%)
- Lack of child’s extracurricular experiences and social interactions	8 (8.79%)	2 (4%)	0 (0%)	6 (27.27%)
Psychological and Relational Difficulties	34 (37.36%)	12 (24%)	8 (42.10%)	14 (63.64%)
- Stress and negative feelings	20 (21.98%)	4 (8%)	7.(36.84%)	9.(40.91%)
- Concern for the child’s well-being	11 (12.09%)	3 (6%)	2 (10.53%)	6 (27.27%)
- Inadequacy and sense of helplessness	11 (12.09%)	6 (12%)	2 (10,53%)	3 (13.64%)

## Data Availability

Data available on request to the last author.

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
