# Peer review of "Parenting a Child with a Neurodevelopmental Disorder during the Early Stage of the COVID-19 Pandemic: Quantitative and Qualitative Cross-Cultural Findings"

_ijerph, 2022, doi:10.3390/ijerph20010499_

Round 1

Reviewer 1 Report

The article is interesting as it is a cross-cultural study. The main question of this paper is about how the magnitude of the perceived changes in 19 child externalizing behavior, parental stress, and discontinuity of therapy - from before to during 20 COVID-19 - affected parental mental health during the pandemic. 

It is a relevant issue in the field, since there is little research in this field, so any research always provides knowledge. It also provides a study in different groups, in a pandemic situation, material.

The sample size can be improved. Due to the sample size, they are not consistent, and for this reason they are only suggested in the conclusions.

As suggestions for improvement, there are two quotes: https://doi.org/10.3390/ijerph19137834 10.3390/ijerph18041474

Regarding the methodology, tables 3 and 4 do not contribute much as the information described is in results. Table 3 is not pertinent, a table of correlations cannot be presented, it is enough to indicate the significant values, and their implication. Please delete table 3.

On the other hand, the qualitative study does not contribute anything, it is better to remove it.

Reviewer 2 Report

Thank you for providing me with the opportunity to review this interesting article. 

Overall, this is a well-written paper. However, I believe that the manuscript could be improved if the following comments were addressed. 

1. Introduction 

 It should be emphasized that the social and cultural context (in this case, that of European countries and the United States) clearly influences parents' experiences. For example, I'd like to learn more about the differences in social policy between European countries and the United States during the pandemic, as well as how each country has dealt with the new circumstances of the pandemic in terms of rehabilitation services and special education for children (and families) with NDD. 

 I'd like to know what the "gold standard" is for treating children with NDD in each country, and how the pandemic has affected the efficacy of these treatments in practice. 

2. Method 

 I'm concerned about the small sample size because it's inconvenient. For example, the total number of participants is 143, with only 31 from the United States. This sample size is small and, in my opinion, cannot produce statistically valid results. This is a significant limitation of the current study. Please explain why you choose a small size for conducting this interesting research. 

 Cronbach alpha should be written separately for each country, such as Italy: | Spain: | US:. 

 What are the study's exclusion criteria? Please explain. 

 Please include the exact question that the parents were required to answer (in the qualitative section of the questionnaire).

3. Results 

 In terms of qualitative analysis, I'm curious how cultural factors and parent perceptions differ between parents from the US and Europe. I recommend citing recent qualitative studies from Europe that investigate parents' experiences and challenges when raising a child with a neurodevelopmental disorder such as ASD. For example, authors should consider including additional literature support by citing relevant work from Papadopoulos (2021), a qualitative study that found that mothers of children with ASD experienced significant burden, distress, and vulnerability, as well as financial and social concerns associated with parenting a child with an NDD. 

4. Discussion 

 What are the study's limitations? 

 Please describe the implications for future research, as well as clinical practice recommendations. 

I hope these suggestions will prove helpful to the authors. I wish the authors the very best with their revisions. 

References

Papadopoulos, D. (2021). Mothers’ Experiences and Challenges Raising a Child with Autism Spectrum Disorder: A Qualitative Study. Brain Sciences, 11, 309. doi:10.3390/brainsci11030309 

Reviewer 3 Report

This is a well-written, clearly conceptualized and, quantitively, a meticulously analyzed study comparing the effect of quarantine related to COVID-19 on children with NDD regarding the stress felt by their parents in having services for their children curtailed and spending so much time with them. What is novel about this study is the comparison is done in relation to three countries—Italy, Spain and the USA. The research is hampered by a poor review of the literature as the majority of the citations are to research that is too old in the field to be relevant currently. As well, some additional details are required regarding the choices made in conducting the quantitative analysis. Furthermore, there is too little information provided concerning the coding of the qualitative analysis across the three different countries for it to be repeatable. Nevertheless, these problems are easily remedied, as is the need to redo the references in relation to the journal’s style. Otherwise, this appears to be fine research.

Line by line suggested edits

1-4 Please eliminate the quotation marks around the title.

37 Please provide a more recent reference to support this claim.

38 Citation 3 is to a reference from 2005; a more recent reference should support this claim.

39 Citation 4 is to a reference that is incomplete and doesn’t identify the year of publication. Using the link, the reference is found to be from 2004 and is too old to support the claim. Citation 5 is to a reference from 2006; the claim needs a more recent reference.

40 Citations 6 and 7 are both to references that are too old to support the claim. Please find more recent references.

42 As mentioned for line 37, citation 1 needs to be updated to more recent research.

44 Both citations 9 and 10 are too old to act as support for this claim. Please find more recent references.

50 Citation 11 is to a reference that is too old. Please find more recent supporting research for this claim.

56 Citation 15 is to a reference from 2016. It is preferable that cited research be within in the last five years. Please find a more recent reference.

57 More recent research is required to support this claim than the research of citation 16.

73 Citation 20 is to an incomplete reference. Please complete this reference.

110 Citation 22 is not to a reference regarding the GHQ-12. Please provide a reference to this questionnaire and references supporting the claim that it is widely-used.

115 Please separate the two citations with a comma and no space. It is likely that citations 22 and 24 are mixed up and reference 24 should be 22 and the reference for 24 should be 22.

127 Why was the PSS selected? What recent research demonstrates it is effective?

132 Citation 27 is to research from 1997. This is far too old to be relevant. Furthermore, the properties revealed were for the full version of the PSS and the current study used a brief scale. Please explain why the full version was not used and provide recent research to support this decision.

156 Please explain why these three tests were considered the appropriate ones to apply in doing the statistical analysis. What would be lost if each of these analyses were not performed.

174 Please provide a reference for the SPSS Package 25.0 and indicate why this package was chosen for the analysis.

176 Please prove more description here regarding the exact question asked in each of the three countries. Was there a difference in what was asked in each of the three countries?

224 Given that this study was conducted in three different countries, how were the qualitative results analyzed for consistency? What procedure was used to compare the results to ensure that the coding of the answers was consistent across each of the different countries.

226 If citations 32 and 33 are to references that are seminal related to qualitative content analysis, this needs to be stated. Otherwise, the research cited is too old. Recent research will be needed to support the use of this type of analysis.

248 The concern here is similar to that related to line 226.

272 As mentioned in relation to line 50, citation 11 is to research that is too old to be relevant. This is even more so regarding reference 34 and 35. Please find more recent references to support your claim.

278 Only reference 36 is recent. As mentioned with respect to line 40, more recent references are required for both citations 6 and 7.

306 Citation 38 is to research that is too old. Please find a more recent reference.

References:

Please see the style guide for MDPI references. All the references need to be redone as per that guide.
